# Learning Latent Causal Semantics from Text: An Empirical Study of Next-Token Predictors Trained on Programs

## Abstract

We present evidence that language models can learn to represent the semantics latent in text despite being trained only to perform next token prediction. Specifically, we train a Transformer model on a synthetic corpus of programs written in a domain-specific language for navigating 2D grid world environments. Each program in the corpus is preceded by a (partial) specification in the form of (textual) input-output examples, and hence the semantics of the programming language enter as a *latent causal variable* in the data generation process. We then probe the trained model's hidden states as it generates a program given a specification. Despite providing no inductive bias toward learning the semantics of the programming language, we find that a linear probe is able to extract abstractions of the program states from the model states, which suggests the model acquires an emergent ability to *interpret* programs in the formal sense. Moreover, there is a strong, statistically significant correlation between the accuracy of the probe and the model's ability to generate a program that correctly implements the specification. To evaluate whether the semantics are represented in the model states rather than learned by the probe, we propose a causal framework for analyzing the effects of probing, and perform interventional experiments that allow us to precisely attribute the accuracy of the probe to the semantics latent in the model's training data (rather than, e.g., the signal used to supervise the probe). In summary, this paper does not propose any new techniques for training language models, but develops an empirical framework for and provides insights into the acquisition and representation of semantics in language models.

## 1 Introduction

Despite the rapidly improving performance of large, pretrained language models (LMs) in a range of downstream tasks, a major open question is whether such LMs capture any semantically meaningful information about the text that they consume and generate (Mitchell & Krakauer, 2023). Empirically, some recent research has found that LMs are insensitive to the semantics of the presented prompts (Webson & Pavlick, 2022; Min et al., 2022; Kavumba et al., 2022), though it has also been observed that these phenomena can diminish with scale (Wei et al., 2023).

It is an open question whether there exists a fundamental barrier which prevents LMs from developing an understanding of language grounded in the semantics of the underlying domain. A common theme is the disconnect between *form* (i.e., text) and *semantics*, which is presumed to lie beneath the surface of form and is therefore inaccessible to an LM trained purely on text (Bender & Koller, 2020). For instance, Browning & LeCun (2022) claim that an LM trained only on text is doomed to "shallow" as opposed to "deep" understanding, while Chomsky et al. (2023) argue that as LMs are trained only to model *correlations* between surface tokens, they cannot grasp the *causal* mechanisms expressed in the text. Indeed, a recent meta-survey reveals a sharp divide within the NLP community, with 51% of respondents agreeing to the statement, "Some generative model trained only on text, given enough data and computational resources, could understand natural language in some non-trivial sense" (Michael et al., 2022).

We present an *empirical* framework for studying the extent to which semantics can emerge in LMs trained solely to perform next token prediction on text. Specifically, our goal is to experimentally evaluate (a formal version of) the following main hypothesis (**MH**):

**Main Hypothesis (informal).** *Semantics cannot be learned from form, i.e., an LM trained only to perform next token prediction on text cannot acquire semantics of the underlying domain.*

Central to our experimental setting is the idea of a *latent causal semantics*—semantics which are not directly observed in the training data, but nonetheless affect the distribution of tokens. Indeed, it follows immediately from our definitions that any latent semantics which is *not* causal in our model cannot be learned from text.

To evaluate whether latent causal semantics can be learned from text, we apply language modeling to the task of *program synthesis*, or synthesizing a program given a (partial) specification in the form of input-output examples. Our primary motivation in adopting this setting is that the semantics of programming languages are well understood and yield rigorous ways of defining the semantics (and correctness) of a program. The training data consists of a corpus of programs and their corresponding specifications, where the formal semantics of the programming language constrains the corpus to programs which correctly implement the input-output examples (and hence constitute a *latent causal variable* in the data generation process).

To evaluate whether the LM has learned semantics, we train a series of small probing classifiers to predict a representation of the program semantics from the LM's hidden states. We find the probe's ability to extract semantics is random at initialization, then undergoes a phase transition during training, with the phase transition strongly correlated with the LM's ability to generate a correct program in response to previously unseen input-output examples. We also present results from a novel interventional experiment. These results indicate that the semantics are represented in the model states (rather than learned by the probe).

Our contributions are as follows:

1. We present a formal model of semantics acquisition in language modeling. We use this model to derive an empirically testable version of **MH**.

2. We present experimental results that support the emergence of semantically meaningful representations in LMs trained to perform next token prediction (Section 3). In particular, we use the trained LM to generate programs given input-output examples, then train probes to extract information about the program state from the model state. We find that the internal representations contain encodings of (1) an abstract semantics—specifically, an *abstract interpretation*—that track the specified inputs through the execution of the program, and (2) predictions of *future* program states corresponding to program tokens that have yet to be generated. During training, these linear representations of semantics develop in lockstep with the LM's ability to generate correct programs across training steps.

3. We design and evaluate a novel interventional technique that enables us to separate the contributions of the LM and probe when extracting semantics from representations (Section 4). Specifically, this technique makes it possible to determine whether (1) the LM representations contain purely (syntactic) transcripts while the probe learns to interpret the transcript to infer meaning, or conversely (2) the LM representations contain semantic state, with the probe simply extracting the meaning from the semantic state. The results indicate that the LM representations are, in fact, aligned with the original semantics (rather than just encoding some lexical and syntactic content), which—together with the results in Section 3—rejects a strong version of **MH**.

More broadly, we present a framework for conducting empirical research on LMs based on the semantics of programming languages. Working with programs allows us to define, measure, and experiment with concepts from the precise formal semantics of the underlying programming language, ideally yielding novel insights that contribute toward a principled understanding of the capabilities of current LMs. Going forward, we believe methods similar to those developed in the present work can offer a complementary formal perspective on how key concepts related to language and cognition can be mapped to the setting of LMs and, more generally, machine intelligence.

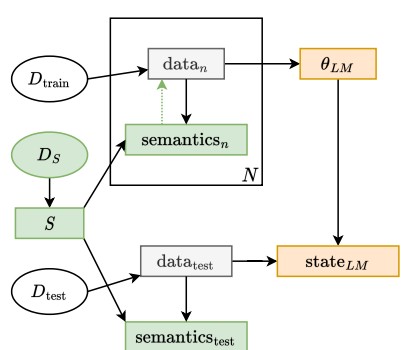

(a) An SCM describing a data generation process with a latent causal semantics.

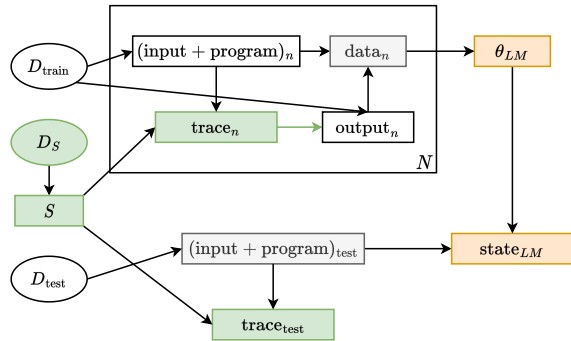

(b) The SCM specialized to our experimental setting, where the textual data consist of programs and input-output examples, and the semantics are given by traces of the program execution.

Figure 1: An illustration of latent causal semantics. In Figure 1a, $D_{train}$ and $D_{test}$ are distributions over textual data (strings). The parameters of the language model $\theta_{LM}$ are trained on a corpus of data sampled from $D_{train}$. At test time, an input prompt $data_{test}$ is sampled from $D_{test}$, and the trained LM produces $state_{LM}$. $S$ is a map which assigns semantics to data; the green dotted line indicates the key causal link between the semantics and the training data, i.e., the text in the training corpus is presumed to be meaningful. Note that the causal link is represented as a cycle here only for brevity; it is removed in the full SCM in the Appendix. The SCM implies that, because the semantics $S$ is shared across both the training and test data, then $state_{LM}$ may no longer be independent of $semantics_{test}$, even if $data_{test}$ is known; in other words, *the output of the LM can capture semantics beyond what is expressed directly in the text of the input*. Figure 1b show how to map this SCM to our experimental domain of programs and their formal semantics.

## 2 BACKGROUND AND SETTING

We begin by introducing the notion of a *latent causal semantics*. Figure 1 expresses the key concepts as a structural causal model (SCM), which describes a data generation process using a directed acyclic graph (Pearl et al., 2000): the nodes represent random variables and the edges indicate the direction of causality. Note that the collection of variables representing semantics (i.e., both the semantic map $S$ and the collection $\{semantics_n\}_{n=1}^{N}$) in Figure 1a constitute latent causal variables in the data generation process: they are *latent* because only the training data $\{data_n\}_{n=1}^{N}$ is observed (and hence the learned parameters $\theta_{LM}$ are independent of the semantics given the training data), [1] and *casual* due to the presence of the green causal link influencing the distribution of text in the training data; we refer to the caption for more detail.

The causal link plays a critical role in the SCM as severing it renders $state_{LM}$ formally independent of $semantics_{test}$ given $data_{test}$, i.e., we have no hope of extracting any information about the semantics of the input from the output of the LM beyond what is already observed in the surface forms. Moreover, the presence of this causal link also yields empirically testable implications which correspond to a formal statement of **MH**. Specifically, propose to study the following two hypotheses:

**Main Hypotheses (formal).** *Let $\theta_{LM}$ in Figure 1a be trained using next token prediction. Then*

$$state_{LM} \perp semantics_{test} \mid data_{test}, \tag{SH}$$

*regardless of whether a causal link is present, and furthermore,*

$$state_{LM} \perp semantics_{test} \mid data_{test}, \{data_n\}_{n=1}^{N}. \tag{WH}$$

---

[1] In fact, *all* semantics are formally latent by construction in our SCM (cf. Bender & Koller (2020), who use this as a basis for their argument that meaning cannot be learned from form, but do not discuss the possibility of a causal link). For instance, consider training on a dictionary consisting of "The definition of $X$ is $Y$", where $S$ maps $X$ to $Y$. In this case, the semantics are functionally observable. Although this would appear to render both **SH** and **WH** vacuous, Berglund et al. (2023) surprisingly find that an LM fine-tuned on a corpus of "$X$ is $Y$" can fail to generalize to the (marginally) more complex semantics "$Y$ is $X$" (which can be interpreted as evidence in favor of **SH**)!

The strong hypothesis (**SH**) is a claim about a fundamental limitation of LMs trained using next token prediction, i.e., even though the SCM does not imply that $\text{state}_{LM} \perp \text{semantics}_{\text{test}} \mid \text{input}_{\text{test}}$, the learning process precludes the LM from accessing the latent semantics at test time. The weak hypothesis (**WH**) hedges this claim by further conditioning on the observed training corpus, the implication being that the LM cannot access the semantics of its inputs beyond what is observed in the surface forms of the input *and its training data*. As this conditional independence *is* directly implied by the SCM, rejecting this hypothesis would actually refute the SCM itself. Although our experiments will be mostly tailored toward evaluating **SH**, we discuss some possible implications of our experiments for **WH** in Section 5, and leave a separate empirical evaluation to future work.

Finally, we remark that Figure 1a models the semantics as an abstract map from data (documents, or input strings) to a space of arbitrary semantics; these could physical states in a system, mathematical objects as in denotational semantics, or the mental states of speakers. This approach is consistent established theories commonly found in both linguistics (Montague et al., 1970; Montague, 1973) and programming languages (Winskel, 1993). Practically speaking, formulating **SH** and **WH** in terms of the general relationship between semantics and form also gives us the freedom to select a domain amenable to rigorous experimentation. The remainder of this section introduces our chosen instantiation of the SCM in Figure 1b: programs ($D_{train}$, $D_{test}$) and their formal semantics ($S$), a domain with precedence in prior theoretical work exploring the relationship between form and semantics in LMs (Bender & Koller, 2020; Merrill et al., 2021).

## 2.1 PROGRAM TRACING AS MEANING

A foundational topic in the theory of programming languages, formal semantics (Winskel, 1993) is the study of how to formally assign meaning to strings in the language. In this work, our model of semantics consists of *tracing* a program's execution: given a set of inputs (i.e, assignments to variables), the trace is the sequence of intermediate values generated as the program executes on the inputs. More generally, Cousot (2002) proved that the ability to trace a program can be used to define a denotational semantics, where each program expression is mapped to a *denotation*—a mathematical object (such as a number or a function)—that precisely describes its meaning. As the meaning of a program can be formally defined by the collection of all its traces, being able to trace the program on a subset of inputs reflects some understanding of the program's meaning in a precise sense.

Beyond this formal perspective, tracing is attractive as a model of program meaning for several reasons. In novice programmers, the ability to accurate trace a piece a code has been directly linked to the ability to explain the code (Lopez et al., 2008; Lister et al., 2009), and computer science education has emphasized tracing as a method of developing program understanding (Hertz & Jump, 2013) and localizing reasoning errors (Sorva, 2013). Expert programmers also rely on tracing, both as a mental process (Letovsky, 1987) and as implemented in the vast array of trace-based debuggers.

**Abstract interpretation**    Given a program semantics, *abstract interpretation* (Cousot & Cousot, 1977) is one way to coarsen the semantics while preserving its compositional structure. For instance, given the multiplication operator $\times$ over the integers $\mathbb{Z}$, we could define an abstract interpretation $\alpha$ by mapping each integer to its sign $\alpha : \mathbb{Z} \mapsto \{-, 0, +\}$, with the corresponding abstract operator $\times^{\alpha}$ defined in the natural way. This abstraction is *precise* because, for any two integers $x, y \in \mathbb{Z}$, we have that $\alpha(x \times y) = \alpha(x) \times^{\alpha} \alpha(y)$ (i.e., $\alpha$ is a *homomorphism*). We leverage abstract interpretation to precisely isolate a subset of the trace semantics.

## 2.2 LANGUAGE MODELING TASK AND TRAINING

**Karel**    Karel is an educational programming language (Pattis, 1994) developed at Stanford in the 1970s, which is still in use in their introductory programming course today (Piech & Roberts, January 2019; CS106A, 2023). The domain features a robot (named Karel) navigating a grid world with obstacles while leaving and picking up markers. Since being introduced by Devlin et al. (2017), Karel has been adopted by the program synthesis community as a standard benchmark (Bunel et al., 2018; Shin et al., 2018; Sun et al., 2018; Chen et al., 2019; 2021b), in which input-output examples are provided, and the task is to produce a program which maps the inputs to the outputs.

Figure 2 gives an overview of our domain. Each 8x8 grid world contains 4 types of tokens: the robot controlled by the program, which is represented by an arrow indicating the direction the robot

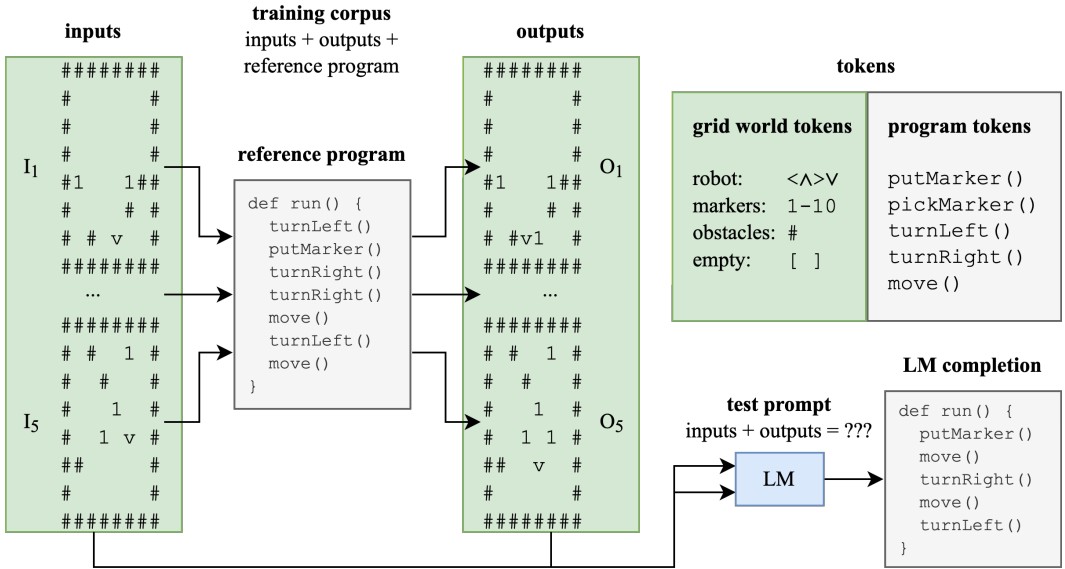

Figure 2: An overview of the Karel domain. We construct training examples by sampling a random reference program, then sampling 5 random inputs and executing the program to obtain the corresponding 5 outputs. The LM is trained to perform next token prediction on a corpus of examples consisting of the concatenation of interleaved inputs and outputs, and finally the program. At test time, we provide only the input-output prefix to the LM, and use greedy decoding to generate a program. The figure depicts an actual reference program and completion from the final trained LM.

currently faces ($\wedge, <, \vee, >$); markers (a space can accumulate up to 10 markers); obstacles (#); or an empty space. We focus on the subset of the language consisting of straight line programs composed from the following 5 operations: `move` advances the robot by one space in the facing direction if there is not an obstacle ahead (otherwise, the robot does not move); `turnRight` and `turnLeft` turn the robot right and left, respectively; `putMarker` and `pickMarker` increment and decrement the number of markers on the space occupied by the robot (with no effect if there are 10 and 0 markers), respectively. Note that the robot obscures the number of markers on the space it currently occupies, and the obscured markers have no effect on the correctness of a program.

**Karel synthetic dataset construction**   Our training set consists of 500,000 randomly sampled Karel programs of lengths between 6 and 10, inclusive. For each program, we randomly sample 5 grid worlds to serve as input, then evaluate the output of the program on each input. We create textual representations for Karel grid worlds by scanning the grid in row order, with one token per grid space. Each training sample consists of the concatenation of the input-output examples (the *specification*), followed by the *reference program*. Note that (1) the training set consists only of programs which are correct with respect to their specification and (2) the intermediate states of the trace are not observed in the training data (hence the traces constitute latent causal semantics). We also generate a test set of 5000 specifications in the same manner, except that the lengths of the sampled reference programs range between 1 and 10. At test time, we consider any program that satisfies the input-output examples to be correct (not just the reference program).

**Training an LM to synthesize programs**   We train an off-the-shelf[2] Transformer (Vaswani et al., 2017) to perform next token prediction on our dataset. To measure synthesis accuracy, we use the LM to generate text starting from a specification using greedy decoding. The completion is correct if it is a well-formed program that maps each input in the specification to its corresponding output. We refer to this as the **generative accuracy** of the LM. The LM achieves a maximum generative accuracy of 92.4% on the test set at step 76000 (out of 80000 total steps, measured every 2000 steps).

---

[2]Specifically, we train a 350M parameter variant of the CodeGen architecture (Nijkamp et al., 2023) in the HuggingFace Transformers library (Wolf et al., 2020) from initialization.

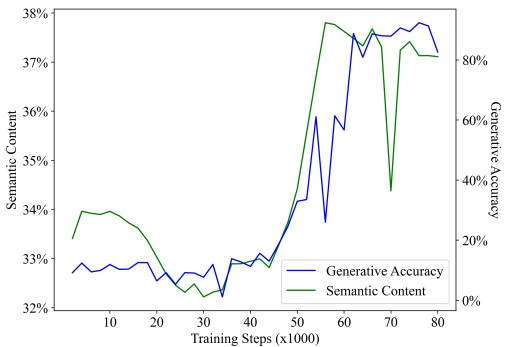

(a) Measuring semantic content with a linear probe.

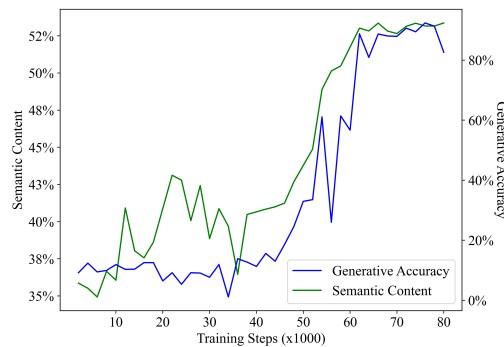

(b) Measuring semantic content with a 1-layer probe.

Figure 4: Comparing the semantic content measured by different probes with the generative accuracy of the LM across training.

## 3 EMERGENCE OF SEMANTICS

We investigate the hypothesis that representations of semantics emerge in the LM state as a byproduct of training the LM to perform next token prediction. Given that the LM achieves a generative accuracy of 92.4%, rejecting this hypothesis would be consistent with **MH**, namely, that the LM has learned to "only" leverage correlations in form of the specifications to consistently generate correct programs.

To test this hypothesis, we conduct a series of probing experiments to extract the direction of the robot from the LM state as 5 separate 4-way classification tasks. The idea is to prompt the LM to generate a program given some inputs, and check whether the LM states contain a representation of the *intermediate program states as it generates the program*.

Every 2000 steps during training, we use the LM to process strings consisting of $input_{test} := (input_0, input_0, \ldots, input_4, input_4, program)$, where the inputs and programs are sampled independently; we obscure the outputs in the specification so as to not "leak" information about the semantics in the prompt, and we duplicate the inputs in the prompt to better match the distribution of tokens seen during training. We then take a snapshot of (1) the final layer hidden states of the LM as it processes each token of the reference program, and (2) the corresponding program states after evaluating the partial program on each of the 5 specified inputs. Figure 3 illustrates this process.

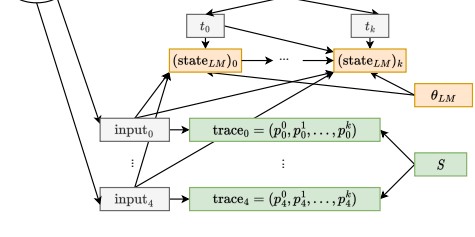

Figure 3: Trace dataset construction: 5 inputs and a program are sampled independently, and a program of length $k$ yields $k$ LM states and 5 execution traces consisting of $k$ program states each.

We repeat this process for each of the training and test sets, producing two *trace datasets* consisting of aligned pairs of LM and program states. We then fit a linear classifier and a 1-layer MLP to predict the direction of the robot given the LM state as input, and evaluate the accuracy of the probes on the test split of the trace dataset. As the facing direction yields a precise abstraction of the full trace semantics, the ability to trace the direction of the robot formally reflects an ability to access an aspect of the program's semantics in a precise sense. In the main text, we refer to the accuracy of a probe to extract the direction from the LM states as the **semantic content** of the LM; additional results for additional semantic features of the program state, such as how far the robot has moved from the initial position and whether the robot is facing an obstacle are contained in the appendix (due to the conclusions being identical).

**Emergence of semantics is correlated with generative accuracy** Figure 4 plots our main results. Our first observation is that the semantic content starts around 33% for both probes (which is close

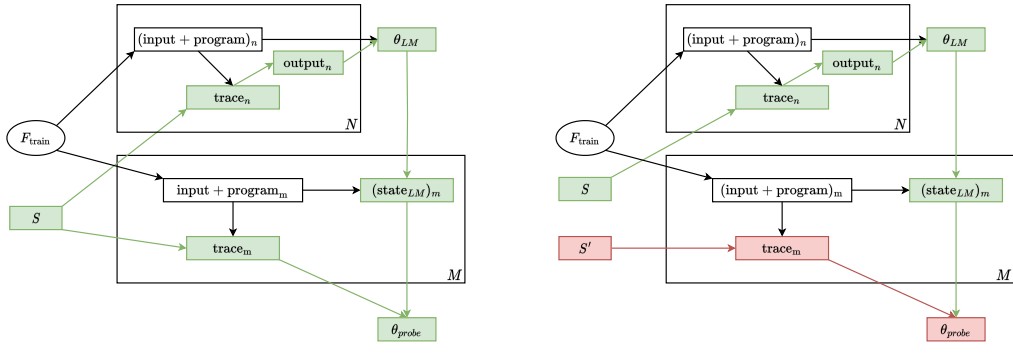

(a) The probe training process.     (b) The proposed interventional experiment.

Figure 5: SCMs describing how supervising the probe has a confounding effect on the observed semantic content (left) and an intervention on the semantics used to supervise the probe (right).

to the baseline performance of random guessing at 25%), and increases over the course of training. This result suggest that the LM states do in fact contain encodings of the semantics, and crucially this semantics emerges within an LM trained purely to perform next token prediction on text. Linearly regressing generative accuracy against semantic content yields a surprisingly strong, statistically significant linear correlation across training steps ($R^2 = 0.756$ and $0.861$ for the linear and 1-layer MLP probes, respectively, and $p < 0.001$), i.e., the variability in the LM's ability to synthesize correct programs is almost completely explained by the semantic content of the LM's state. This suggests that, within the scope of our experimental setup, learning to model the distribution of correct programs is directly related to learning the semantics of programs.

## 4 ATTRIBUTING SEMANTIC CONTENT TO MODEL STATES (NOT THE PROBE)

In this section, we address a central challenge of drawing conclusions from probing classifiers, namely, that a high semantic content could be due to either meaning being represented in the model states *or* the probe learning the task itself (Hewitt & Liang, 2019; Belinkov, 2022). In our case, because the probe is explicit supervised on the semantics $S$, this introduces the two paths in Figure 5a from $S$ to the probe which could influence the measured semantic content. For instance, the model states may simply encode the inputs and a list of tokens in the program generated thus far, while the probe reads off then interprets the tokens one-by-one. This problem renders even the question of how to formally interpret the result of a probing experiment an open question (Pimentel et al., 2020).

Instead, we propose a novel intervention study that can be used to establish a baseline for empirically validating **SH**. Specifically, we conduct the causal intervention described in Figure 5b to isolate the contribution of the probe: we define an **alternative semantics** $S'$ by replacing the semantics of individual operations in the language with a different operator. Then, we retrace the program according to the alternative semantics and train a new probe to decode the *original* model states to the *alternative* semantic states.

Formally, to reject **SH** we need to show that

$$P(\text{semantics}_\text{test}|\text{data}_\text{test}, \text{state}_{LM}) > P(\text{semantics}_\text{test}|\text{data}_\text{test}), \qquad (1)$$

since this would violate the independence of the LM state on the semantics given the text. However, our construction of data$_\text{test}$ consists only programs and inputs sampled *independently* of the semantics $S$. Thus, this requirement reduces to showing that

$$P(\text{semantics}_\text{test}|\text{state}_{LM}) > P(\text{semantics}_\text{test}) \qquad (2)$$

However, as the left-hand side is intractable to measure in general, we instead estimate

$$P(\text{semantics}_\text{test}|\text{probe}(\text{state}_{LM})) > ? \qquad (3)$$

using the learned probe, where rejecting **SH** on the basis of this estimate now requires the counterfactual claim that *if* the LM state had been independent of the semantics (given the data), *then* the probe would have extracted even less information.

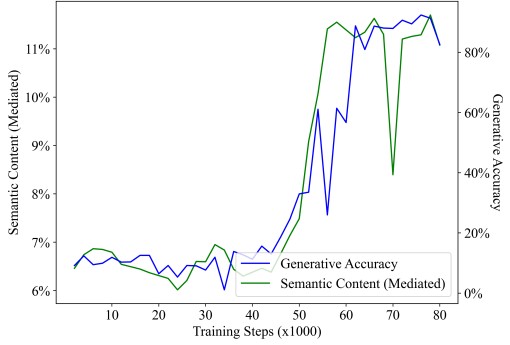 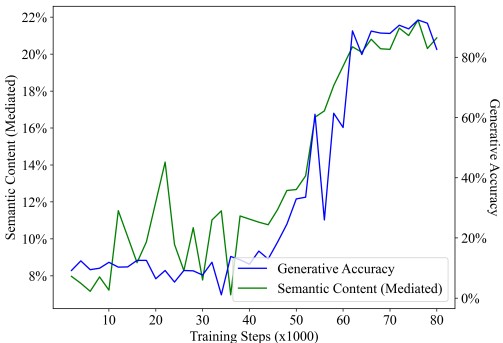

(a) Measuring adjusted semantic content with a linear probe.

(b) Measuring adjusted semantic content with a 1-layer probe.

Figure 6: Comparing the semantic content adjusted by an alternative semantics baseline.

Under a key assumption that there are no additional confounders between the choice of $S$ and $S'$ and the probe, this counterfactual can be converted into an empirical question of whether one can find an $S'$ which reduces the semantic content below what was observed under $S$. As such, the experiment relies crucially on identifying an alternative semantics which is as different as possible from the original semantics, without posing any additional challenges for the probe. In our setting, we claim that this assumption is satisfied by preserving the set of operators (as opposed to inventing completely new operations, e.g., move two spaces in one step). Concretely, we define the alternative semantics $S'$ as follows:

| original | `pickMarker` | `putMarker` | `turnRight` | `turnLeft` | `move` |
|---|---|---|---|---|---|
| alternative | `turnRight` | `turnLeft` | `move` | `turnRight` | `turnLeft` |

For instance, the `turnRight` operation in the original semantics would have the robot turn 90 degrees clockwise, but in the alternative semantics the robot instead advances by a step (i.e., according to the original definition of the `move` operation).

Figure 6 displays the results of this experiment, where we supervised probes to predict the direction of the robot according to the alternative semantics $S'$, then plot the adjusted semantic content:

$$\text{SemanticContent}(\theta_{LM}(S), \theta_{probe}(S)) - \text{SemanticContent}(\theta_{LM}(S), \theta_{probe}(S'))$$

(the notation $\theta(S)$ refers to the parameters fit on the semantics $S$). We observe that adjusted semantic content is significantly positive, which suggests that a significant portion of the observed semantic content can be attributed to the LM states. Furthermore, the adjusted semantic content increases over the course of training, and regressing against the generative accuracy yields *higher* $R^2$ than the unadjusted semantic contents ($R^2 = 0.821$ and $0.883$ for the linear and 1-layer MLP probes, respectively, and $p < 0.001$), which we attributed to smoothing out noise in the early stages of training. We thus conclude that the LM does, in fact, acquire semantics over the course of training, despite being trained only via next token prediction on text.

## 5 DISCUSSION AND RELATED WORK

**Meaningful representations in LMs**   Li et al. (2023) train a Transformer on transcripts of Othello, then probe the model activations (not the hidden states) to extract the board state. Li et al. (2021) fine-tune several pretrained LMs on text that describes evolving situations, then probe the model states to test propositions about entities in the situation. Abdou et al. (2021) find that pretrained LMs' representations of color terms are geometrically aligned with CIELAB space.

This work makes several novel contributions within this body of literature. To the best of our knowledge, we are the first to develop a formal model of meaning acquisition in language models via *causal latent semantics*. On the empirical front, we are also first to explore how semantics in

LMs emerges over time (instead of a single snapshot at the end of training), and find a strong, linear relationship between the emergence of semantics and correctness.

**Analyzing the behavior of LMs** Researchers have investigated the ability of LMs to successfully complete a range of semantically meaningful tasks (Austin et al., 2021; Toshniwal et al., 2022; Patel & Pavlick, 2022; Liu et al., 2023). Unlike our research, which probes the internal state of the LM to determine the presence or absence of semantically meaningful state, this line of research works only with the externally observable behavior of the LM.

**Probing** Probing (Shi et al., 2016; Belinkov & Glass, 2019) is widely used as a technique to investigate the inner workings of LMs. A key challenge is controlling for what is learned by the probe rather than latent in the LM (Belinkov, 2022). Hewitt & Liang (2019) develop *control tasks* for word-level properties in the context of probing for parts of speech in LM representations. They compare against the performance of a probe that maps from the model states to a dataset with a *random* part of speech assigned to each word. In our case, the control task approach would assign a random label to each program state; however, this would destroy the compositional structure of the program, and therefore be insufficient as a stand-in for the requisite counterfactual inference. Instead, we establish a baseline by intervening on the semantics of program constructs, and generate a new label for each program state by evaluating the program according to the alternative semantics, while controlling for the complexity of the alternative semantics. Our technique better is thus suited than control tasks when probing for semantic rather than syntactic information (Pimentel et al., 2020).

**Program synthesis with LMs** There is a growing body of work on training large-scale, Transformer-based LMs for program synthesis (Chen et al., 2021a; Li et al., 2022; Nijkamp et al., 2023; Fried et al., 2023; Austin et al., 2021), as well as program synthesis as a benchmark for LMs (Hendrycks et al., 2021; Liang et al., 2022), but none of this previous research investigates the internal representations of LMs for evidence of semantic state. We note that these papers have also observed that the BLEU score with respect to a reference solution is not a good predictor of the LM's competency, which complements our results regarding the LM's perplexity on the training corpus.

**Grounding programs from text** Prior work has argued specifically that LMs cannot ground programs given only textual hints of semantics (Merrill et al., 2021). Bender & Koller (2020) concede that semantics could be learned from programs paired with unit tests, but assert this requires a "learner which has been equipped by its human developer with the ability to identify and interpret unit tests," implying that an LM would require an additional supervised signal to associate unit tests with the semantics of programs. In contrast, our results indicate that an LM learns the semantics of programs from textual instances of input-output behavior using only next token prediction.

**SH and WH** The discussion so far has focused on rejecting **SH**. Indeed, attributing the semantic content of the probe to the state of the LM constitutes strong evidence in favor of refuting **SH** (Section 4). However, addressing **WH** is far more difficult, as it requires conditioning on the entire training corpus. The main argument in favor of rejecting **WH** is that, because the LM is trained only on programs of length 6 or greater, the training data cannot reveal any information about how the traces are assigned to, e.g., the 2nd program state. Accepting this argument however requires rejecting the SCM in Figure 1a. We hypothesize that **WH** can be resolved by the presence of an *unobserved confounder*: as LMs have evolved to efficiently process human language, the architecture of the LLM (auto-regression, parameter sharing, attention) may be biased toward the space of "natural" semantic structures (compositional, causal, acyclic). We leave the exploration of this hypothesis to future work.

## 6 CONCLUSION

The question of whether semantics can be learned from text has garnered considerable interest in recent years. This paper presents empirical support for the position that **semantics is learnable from form**. More broadly, the formal approach to semantics presented here offers a principled foundation for studying semantics in models of language—a question of both practical and philosophical importance.

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

## A  EXPERIMENTAL DETAILS

### A.1  KAREL GRAMMAR SPECIFICATION

We use the same grammar as Devlin et al. (2017), except with loops and conditionals removed.

$$\begin{aligned}
\textbf{Prog } p &:= \texttt{def run():} s \\
\textbf{Stmt } s &:= s_1; s_2 \mid a \\
\textbf{Action } a &:= \texttt{move()} \mid \texttt{turnRight()} \mid \texttt{turnLeft()} \mid \texttt{pickMarker()} \mid \texttt{putMarker()}
\end{aligned}$$

### A.2  FACING DIRECTION ABSTRACTION

The facing direction abstraction maps a program state to the facing direction of the robot within that program state. It follows that the abstraction function $\alpha$ is simply a projection operator that forgets all information about the program state except for the facing direction of the robot. The abstract semantics are given by:

| full semantics | `pickMarker` | `putMarker` | `turnRight` | `turnLeft` | `move` |
|---|---|---|---|---|---|
| abstract semantics | `id` | `id` | `turnRight` | `turnLeft` | `id` |

where `id` is the identity operator that does not affect the facing direction. Clearly, $\alpha$ is a homomorphism, and so the abstraction is exact.

We identify an important property of this choice of $\alpha$, which is relevant to the experimental design. First, note that $\alpha$, being a projection, is linear. Second, recall that the abstraction is exact when the abstract semantics are a subset of the full semantics, in a precise sense. Combining these two facts yields that any conclusions we draw satisfy *soundness* with respect to the full semantics, i.e., we are not looking for anything "extra": if there exists a linear representation of the full semantics, then there exists a linear representation of the abstract semantics; additionally, if the semantic content with respect to the abstract semantics is high, this constitutes evidence that the model has indeed acquired an aspect of the original full semantics. Note that if the abstract semantics were not a subset of the full semantics, then the semantic content may be high due to measuring something which conceptually "falls outside of" (or is unrelated to) the full semantics—in this case, high semantic content with respect to the abstract semantics may not constitute evidence that the model has acquired an aspect the full semantics. Hence, using a *precise* abstraction (as we do) is one way to ensure a positive result is still sufficient grounds for rejecting **MH**.

### A.3  TRAINING AND LANGUAGE MODEL DETAILS

We used the non-pretrained 350M parameter variant of the CodeGen architecture (Nijkamp et al., 2023) from the HuggingFace Transformers library (Wolf et al., 2020), implemented in PyTorch (Paszke et al., 2019). We used the AdamW optimizer (Loshchilov & Hutter, 2019) (but no weight decay), a learning rate of 5e-5, a block size of 2048, and a batch size of 16. All program and grid world tokens are represented by special tokens, and the embeddings are trained from scratch. We trained for 80000 steps. Using a single NVIDIA A100 GPU with 80GB of VRAM, training the LM takes around 8 days.

The probe consists of a layer normalization followed by a single linear layer. Note that the hidden states of the CodeGen architecture are passed through a layer normalization as the final layer, so we just re-normalize after average pooling the hidden states. The training set is formed from the first 100000 aligned traces in the training trace dataset. We train for a total of 100 epochs using the AdamW optimizer with a weight decay of 1e-4, a learning rate of 0.01 that decays by .1 at 75 and 90 epochs, and a batch size of 256. Using a single NVIDIA A100 GPU, training each probe takes around 30 seconds.

| Length | Accuracy |
|--------|----------|
| 1 | 99.8% |
| 2 | 99.1% |
| 3 | 99.2% |
| 4 | 98.5% |
| 5 | 97.3% |
| 6 | 96.4% |
| 7 | 94.8% |
| 8 | 92.9% |
| 8 | 89.5% |
| 8 | 86.3% |

Table 1: The generative accuracy of the final trained LM on the test set, separated by the length of the reference program used to generate the specification.

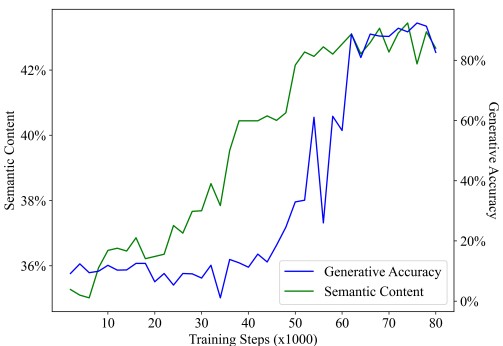
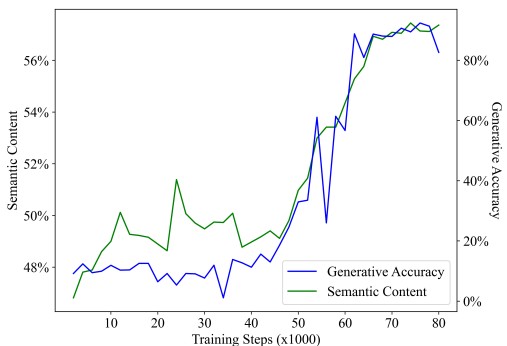

(a) Measuring semantic content with a linear probe.   (b) Measuring semantic content with a 1-layer probe.

Figure 7: Semantic content of probing for the position of the robot as an offset of the robot from its starting position.

## B ADDITIONAL EXPERIMENTAL RESULTS

### B.1 RESULTS BY PROGRAM LENGTH

This section presents results which demonstrate that the LM learns to synthesize correct programs for specifications generated by reference programs of up to length 10, including for lengths *shorter* than what it has seen in its training data.

Table 1 displays the results of this analysis. We see that the LM is able to accurate generate programs that satisfy the specifications across all lengths, with only a moderate drop in accuracy as the reference program length approaches the maximum length of 10.

### B.2 ADDITIONAL TASKS

We additional probe the LM states for two additional features of the semantics: the position of the robot (Figures 7 and 8) and whether the space in front of the robot is clear when a `move()` instruction is issued (Figures 9 and 10). Note that the second property tests whether the LM can acquire the semantics of a conditional.

## C FURTHER DISCUSSIONS OF RELATED WORK

Austin et al. (2021) evaluate a 137 billion parameter LM trained on a mixture of natural language and programs to synthesize Python programs given a natural language description and three input-output assertions. They find that sampling 80 programs from the LM yields at least one accurate program on

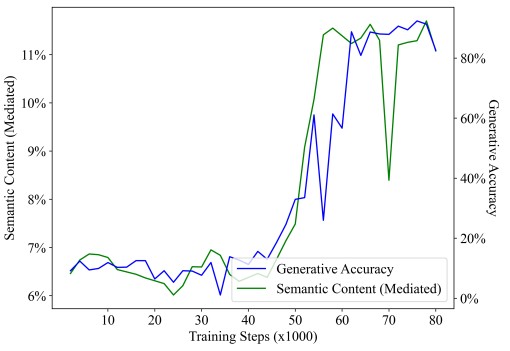 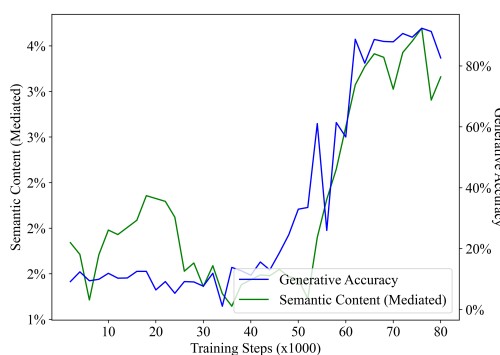

(a) Measuring adjusted semantic content with a linear probe.

(b) Measuring adjusted semantic content with a 1-layer probe.

Figure 8: Adjusted semantic content of probing for the position of the robot as an offset of the robot from its starting position.

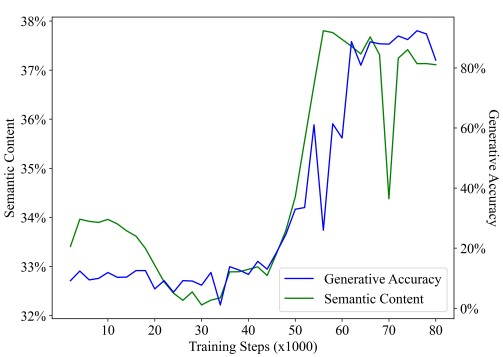 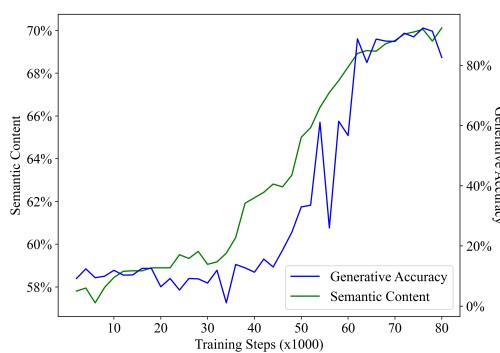

(a) Measuring semantic content with a linear probe.

(b) Measuring semantic content with a 1-layer probe.

Figure 9: Semantic content of probing for whether the space in front of the robot is clear, when a `move()` operation is generated.

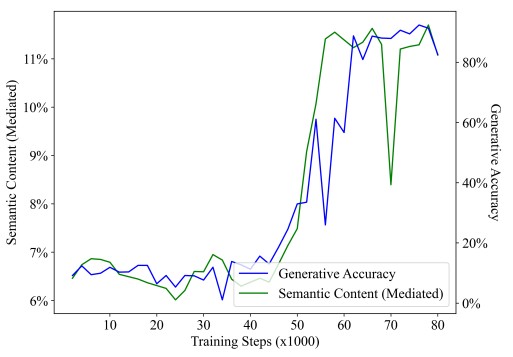 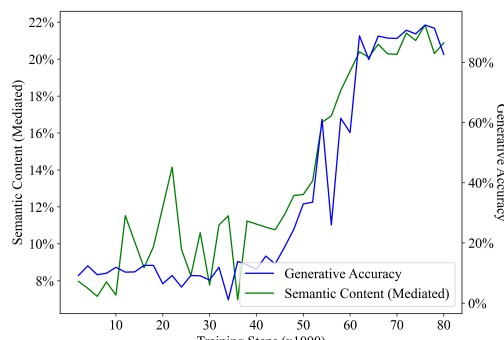

(a) Measuring adjusted semantic content with a linear probe.

(b) Measuring adjusted semantic content with a 1-layer probe.

Figure 10: Adjusted semantic content of probing for whether the space in front of the robot is clear, when a `move()` operation is generated.

around 60% of the tasks, but the LM is only able to generate the output of the program 29% of the time when using greedy decoding. They conclude that LMs do not learn any substantial amount of semantics, despite being able to synthesize correct programs.

We offer three possible explanations: (1) the prompts provided to the LM often contain natural language descriptions of the algorithm, and hence their evaluation is closer to translation than pure synthesis; (2) continuing to train the LM beyond 60% accuracy would yield an LM that is better at predicting the output of the program; and (3) we use greedy decoding for both synthesis, whereas top-k or sampling-based metrics may overestimate the proficiency of the LM at synthesis.

