# OpenReview forum: "Learning Latent Causal Semantics from Text: An Empirical Study of Next-Token Predictors Trained on Programs"
_ICLR.cc/2024/Conference — Submitted to ICLR 2024_

### Official Review · Reviewer_G8pC · 2023-10-30

**Soundness:** 3 good
**Presentation:** 2 fair
**Contribution:** 3 good
**Rating:** 5
**Confidence:** 4

**Summary:**

The paper investigates whether language models (LMs) learn to represent
semantics trained only on the next token prediction task. The LM has to
synthesize programs in a subset of the Karel educational programming languages
to move a robot and place markers in a grid world. The semantics are a latent
causal variable in the data generation process.

The authors trained probes to predict the direction of the robot (and other
semantic content) from the LM states. The outputs are not given to the probe to
avoid leaking the semantics. There is a very strong correlation between the
accuracy of the LM and the amount of semantic content the probes can extract.

A recurring question in these probing experiments is whether the semantic
content was learned by the LM or the probe. The authors propose an intervention
study to isolate the contribution of the probe: they change the semantics by
switching the operators of the robot, and assess how well the probe can learn it
from the LM states trained with the original semantics.

**Strengths:**

The paper is one of the firsts to present evidence that LMs learn and represent
the underlying semantics in spite of being trained on the next token prediction
task. A very interesting part is that the development of these representations
correlates with the accuracy of the LM during training.

The paper also provides a causal inference framework that could help further
study of semantics representations in LMs.

**Weaknesses:**

Part of the main contributions of the paper is that the internal representations
contain encodings of future program states. I was looking forward to reading
about this, but I couldn't find it in the paper.

I couldn't find the full SCM in the Appendix (it was stated in the caption of
Figure 1 that it would be there).

I'm not sure that the paper presented a formal model of meaning acquisition in
LMs as it states in Section 5. It did not give insights about how meaning is
acquired: it demonstrated that semantic content is indeed acquired, but the
network remained a black box.

I found the presentation of the problem and the hypotheses (a Main Hypothesis to
be rejected, and the Stronger and Weaker Hypotheses) a little bit
counterintutive. As no statistical tests were actually done I don't see the need
for formulating them this way (as it's the opposite of the thesis of the paper).
Later the paper uses the positive hypothesis (the thesis), e.g., at the
beginning of Section 3: "We investigate the hypothesis that representations of
semantics emerge...".

I think that this presentation made Section 4 hard to follow. Also, to me the
main question seems not whether LMs can learn semantic content at all (like in
Eq (1)), but how well they learn it, and whether that contributes to the
accuracy (like in Figure 4).

Two limitations:
- that although the Karel programming language has selection and
iteration, the presented examples don't have them: they are just sequences
("straight line programs").
- the problem is relatively simple and the transformer used
is small (350M parameters). I don't think that's a problem for a first
investigation of these phenomena.

Small notes:
- page 5: "evaluate the output of the program on each input": I guess the
  program is evaluated on each input to obtain the output?
- typo in Section 5: "better is thus"

**Questions:**

I don't understand the green causal link in Figure 1a. How is it different from
the other causal links? Why are there no such links for the test set and in
Figure 1b?

How different are the generated programs from the reference programs?

There is a drop in Semantic Content on Figure 4a at around 70 000 training
steps. Have you found an explanation for this?

Why do we use and plot (on Figure 6) the adjusted semantic content and not just
the semantic content obtained using S'? Wouldn't showing that that's close to 0
a better elucidation?

---

### Official Review · Reviewer_gauF · 2023-10-30

**Soundness:** 2 fair
**Presentation:** 2 fair
**Contribution:** 2 fair
**Rating:** 5
**Confidence:** 2

**Summary:**

The paper studies the effect of the language model that is trained to predict the next token in a synthetic corpus of programs written in a specific domain-specific language. These programs are accompanied by textual input-output examples, which introduce the semantics of the programming language as a hidden variable. The researchers found that the trained model can extract abstractions of program states despite no explicit bias toward learning the language's semantics.

The study employs a causal framework to analyze probing effects and conducts experiments to attribute the probe's accuracy to the model's latent understanding of the language's semantics. The paper doesn't introduce new training techniques for language models but provides insights into how they acquire and represent semantics.

**Strengths:**

- The paper is carefully written and the motivation is well explained.
- The authors conduct a thorough related-work study.

**Weaknesses:**

- The experiment is not enough. This paper only studies a small example of using a language model for learning latent causal variables.
- The novelty of this work is not enough. This paper claims to propose a formal model of semantics acquisition in language modeling, which seems to be overclaimed and lacks sound verification in the rest of the paper. The authors might consider other more relevant conferences, like human-computer interaction.
- The presented structural causal model is not the common model presented in the  Judea Pearl 2000 paper. Why it must be this format (shown in Figure 1)? Please provide a more solid definition.

**Questions:**

- The definition of "semantics", "latent semantics" and "latent casual semantics" has been discussed throughout history, the author should define those more carefully.

---

### Official Review · Reviewer_9VK4 · 2023-10-31

**Soundness:** 3 good
**Presentation:** 3 good
**Contribution:** 3 good
**Rating:** 5
**Confidence:** 4

**Summary:**

The paper presents a general approach, grounded in Programming Languages (PL) Theory, for understanding the internal behavior and representations of Language Models (LMs). The stated goal of the paper is to be able to answer whether LMs learn underlying semantics of the domain or not. To conduct this investigation, the idea is to use the formal notion of semantics of programming languages. In particular, the authors train an LM to synthesize code in an educational domain-specific language (for navigating grid worlds) given input-output examples, and then design an experiment to probe if the LM "understands" programs using the notion of trace semantics of programming languages. Given a program, they check via probes if the state of the robot (a part of the program state) is correctly represented in the hidden states of the LMs as the LM processes each line of code. It indeed turns out that the correct state of the robot can be recovered from the hidden states. Moreover, a counterfactual analysis also suggests that it is not the probe itself that has learnt to encode the robot state. The overall hypothesis and experiment design are formalized using the framework of structural causal models.

**Strengths:**

1. It is an innovative and potentially fruitful idea to use semantics of PL for understanding behavior of LMs.

2. Framing the questions in a causal language helps make the investigation and experimental design precise.

3. The results from this paper confirm similar empirical findings from prior work and add to the growing body of evidence that LMs indeed learn something deeper about their domains beyond mere syntactic correlations.

4. The finding that semantic understanding evolves in lockstep with LM ability to synthesize correct programs in interesting.

**Weaknesses:**

1. The paper presents interesting ideas but I find the claims to be over-stated. The connection between semantics of PLs and semantics of natural languages is neither formal nor obvious. While it may be reasonable to claim from the experiments that LMs learn some semantic notions of PLs, generalizing to say LMs learn semantics is not well-supported. Moreover past works such as [1] and [2] already provide evidence that LMs learn something deeper than syntactic structure, i.e., refute the main hypothesis, at least in a narrow sense.

2. Although the use of counterfactual analysis to determine the effect of the probe is important, it seems like an interventional analysis as in [1], where the internal representations of the LMs are directly modified as guided by the probe, would provide even stronger evidence.

3. I find the usage of semantics throughout the paper to be imprecise. For instance, what is the formal definition of the semantics map S? What does $semantics_{test}$ formally mean? I also found the presentation in Section 4 to be hand-wavy and hard to follow. For example, what does probability of $semantics_{test}$ mean? In Equation (3), what does it mean to estimate an inequality? Isn't the value of the left-hand side term in the inequality being estimated?

[1] Li, K., Hopkins, A. K., Bau, D., Viégas, F., Pfister, H., & Wattenberg, M. (2022). Emergent world representations: Exploring a sequence model trained on a synthetic task. arXiv preprint arXiv:2210.13382.

[2] Nanda, N., Chan, L., Liberum, T., Smith, J., & Steinhardt, J. (2023). Progress measures for grokking via mechanistic interpretability. arXiv preprint arXiv:2301.05217.

**Questions:**

In additions to the questions in the **Weaknesses** section, I have the following questions:

1.  Why is the distinction between probing hidden states vs activations (as in [1]) important?

2. What is the precise nature of the probe? In particular, do you learn five separate probes, one for each input? Why not just give a single input instead of five? Also, are each of these five inputs further duplicated (as suggested by the definition of $input_{test}$)? If so, why? Is the input given to each probe the same, i.e., all the hidden states of the LM at the final layer? What accuracy is being reported in Figure 4? Is it the average accuracy of the five separate probes?

[1] Li, K., Hopkins, A. K., Bau, D., Viégas, F., Pfister, H., & Wattenberg, M. (2022). Emergent world representations: Exploring a sequence model trained on a synthetic task. arXiv preprint arXiv:2210.13382.

---

### Official Review · Reviewer_hPVi · 2023-10-31

**Soundness:** 2 fair
**Presentation:** 2 fair
**Contribution:** 2 fair
**Rating:** 3
**Confidence:** 4

**Summary:**

This paper tackles a central problem in languages: the emergence of semantics from forms. To understand this, the authors use the task of program tracing as an example and demonstrate that LMs can learn semantics from a standard next-token prediction pretraining task, and further conduct probing analysis.

**Strengths:**

Overall, I think the paper is tackling an undoubtedly important question about to what extent the model can learn semantics from training tasks. In addition, Despite being very concrete, the programming task that the authors proposed is very interesting, and I believe there is much more to exploit.

**Weaknesses:**

The manuscript requires significant revisions before it can be considered for acceptance. Specific areas of concern include:

1. **Scope and Claims of the Study**: The current paper is more aptly described as a case study examining LMs' encoding of semantics and forms. The connection between LMs and their capacity to learn semantics is a broad topic, and the paper's current evidence is insufficient to support its overarching claims. To address this:
  - Reframe the paper's conclusions to be more specific, grounded, and devoid of overgeneralizations (e.g., in the Abstract, Introduction, and Conclusion sections).
  - Enhance the diversity and robustness of your evidence by introducing at least two more diverse examples.

2. **Claims on Causality**: The assertion that LMs inherently understand causality is not sufficiently supported. While ML models can indeed identify latent semantics in supervised tasks—a consensus in both image and language processing—the challenge lies in distinguishing *causality* from mere *correlation*. To me, your experiments appear to train LMs to transition between states using text-encoded actions (programs), a concept already explored in domains like RL and robotics. If the claim is that LMs can learn causal semantics, the same could be said for many other algorithms. Please elucidate the nature of causality in your experiments and justify how your reasoning and experiments demonstrate causality.

3. Probing Study: The probing analysis in your manuscript doesn't yield novel insights. Previous studies have already established that latent embeddings exist in LMs' representation space, and such embeddings can be harnessed in unsupervised ways to achieve commendable results (as evidenced by [1]. BTW, a comprehensive comparison with such works would be beneficial). Given this background, the paper should clarify the unique contributions and findings of the probing performance.

[1] Discovering Latent Knowledge in Language Models Without Supervision

**Questions:**

See the weaknesses above.

---

### Meta-Review · Area_Chair_X4MS · 2023-12-06

**Metareview:**

This paper aims to explore if the next token prediction objective of Transformers leads to an underlying understanding of code semantics. The study is performed in a simple programming language, Karel, where it is easy to ground semantics to state. Results suggest that Transformers acquire some (latent) notion of program semantics within this setting.

## Strengths
- Attempt to explore an interesting question.
- An interesting approach to the question and good use of causality.

## Weaknesses
- The exploration happens only in Karel and Karel programs. It's hard to generalize this to arbitrary code.
- The results/discussion may be overgeneralizing the observations and additional evidence would be useful.

Given the above, and the fact that the authors did not provide any answers to the reviewer comments, this paper cannot be accepted at its current state. Having said that, I would encourage the authors to address the concerns discussed and resubmit.

**Justification For Why Not Higher Score:**

The reviewers raise multiple interesting questions that the authors did not respond to. Without resolving these, it would be hard to accept this paper and ensure its technical correctness.

**Justification For Why Not Lower Score:**

N/A

---

### Decision · Program_Chairs · 2024-01-16

Reject